# Association of Phenotypic Markers of Heat Tolerance with Australian Genomic Estimated Breeding Values and Dairy Cattle Selection Indices

**DOI:** 10.3390/ani13142259

**Published:** 2023-07-10

**Authors:** Richard Osei-Amponsah, Frank R. Dunshea, Brian J. Leury, Archana Abhijith, Surinder S. Chauhan

**Affiliations:** 1School of Agriculture, Food and Ecosystem Sciences, Faculty of Science, The University of Melbourne, Melbourne 3010, Australia; richard.oseiamponsah@unimelb.edu.au (R.O.-A.); fdunshea@unimelb.edu.au (F.R.D.); brianjl@unimelb.edu.au (B.J.L.); apayyanakkal@student.unimelb.edu.au (A.A.); 2Department of Animal Science, School of Agriculture, College of Basic and Applied Sciences, University of Ghana, Legon, Accra P.O. Box LG 226, Ghana; 3Faculty of Biological Sciences, The University of Leeds, Leeds LS2 9JT, UK

**Keywords:** genomic selection, genotyping, heat stress, robotic dairy, selection index

## Abstract

**Simple Summary:**

In Australia, heat waves in the summer are becoming hotter, longer, and more frequent. Heat stress causes physiological and behavioural perturbations in dairy cattle, compromising animal welfare and production. We investigated the relationship between heat-tolerant phenotypes and the genomic estimated breeding values (GEBVs) for Australian economic, productive, and heat tolerance selection indices in a Holstein Friesian lactating dairy cow herd. The study found positive associations between heat-tolerant phenotypes and GEBVs for heat tolerance, feed saved, fertility, and fat percentage. Selection for heat tolerance should ensure the sustainability of production under hot summer conditions.

**Abstract:**

Dairy cattle predicted by genomic breeding values to be heat tolerant are known to have less milk production decline and lower core body temperature increases in response to elevated temperatures. In a study conducted at the University of Melbourne’s Dookie Robotic Dairy Farm during summer, we identified the 20 most heat-susceptible and heat-tolerant cows in a herd of 150 Holstein Friesian lactating cows based on their phenotypic responses (changes in respiration rate, surface body temperature, panting score, and milk production). Hair samples were collected from the tip of the cows’ tails following standard genotyping protocols. The results indicated variation in feed saved and HT genomic estimated breeding values (GEBVs) (*p* ≤ 0.05) across age, indicating a potential for their selection. As expected, the thermotolerant group had higher GEBVs for HT and feed saved but lower values for milk production. In general, younger cows had superior GEBVs for the Balanced Performance Index (BPI) and Australian Selection Index (ASI), whilst older cows were superior in fertility, feed saved (FS), and HT. This study demonstrated highly significant (*p* ≤ 0.001) negative correlations (−0.28 to −0.74) between HT and GEBVs for current Australian dairy cattle selection indices (BPI, ASI, HWI) and significant (*p* ≤ 0.05) positive correlations between HT and GEBVs for traits like FS (0.45) and fertility (0.25). Genomic selection for HT will help improve cow efficiency and sustainability of dairy production under hot summer conditions. However, a more extensive study involving more lactating cows across multiple farms is recommended to confirm the associations between the phenotypic predictors of HT and GEBVs.

## 1. Introduction

Animal agriculture remains an essential source of livelihood, income, and food security, particularly in the developing world [1,2]. In the future, climate change and its negative impact on the quality of feed, water availability, animal and milk production, livestock diseases, animal reproduction, and biodiversity [3,4,5] are expected to worsen, putting the livelihoods of millions at grave risk [6]. In the face of this challenge, climate-smart livestock breeding programmes should be pursued [7]. Research on livestock genomes should help improve heat tolerance (HT) [8], match genotypes with production environments [9], and select breeding stock to ameliorate the effects of heat stress (HS) [10]. Heat stress is the best-characterised stress with severe impacts on reproductive performance in dairy cattle among all the physiological stressors [11]. Heat stress led to USD1 billion in losses annually in the United States dairy industry alone [12] two decades ago and is likely to be more costly today. More recent estimates indicate that HS exposure just during the dry period of the dam is estimated to cause USD810 million in milk losses annually in the United States [13]. Late-gestation HS effects the development of the foetus, reducing daughter survivability and milk production [14]. Furthermore, HS adversely effects the innate and adaptive immune functions of pregnant animals and their offspring, influencing growth rate, morbidity, and mortality [15]. Physiologically, animals have developed coping mechanisms (acclimation, acclimatisation, and adaptation) to minimise the impact of such environmental stressors. Acclimation refers to a coordinated phenotypic response generated by the animal to a specific environmental stressor, while acclimatisation refers to a coordinated response to several simultaneous stressors. Adaptation involves genetic changes as adverse environments persist over several generations of a species [16]. A study of genetic differences between adapted animals provides valuable information on the genes associated with acclimation and acclimatisation [16]. Emerging strategies to improve the heat tolerance (HT) of animals include introgression of thermotolerant genes [17]. Already an HT gene, SLICK, commonly found in heat-resistant cattle (e.g., Senepol, Brahman), has been introduced in dairy breeds, such as the Holstein [18,19]. This gene confers better thermotolerance due to increased thermoregulatory ability, which reduces HS [20]. The effects of the SLICK allele mutation on the physiological responses to HS can be detected in Holstein calves as early as the preweaning stage of life [21]. Genomics comprises a set of valuable technologies implemented as selection tools in dairy cattle commercial breeding programs [22], and genomic selection uses genome-wide DNA markers to capture the effects of many mutations that influence variation in complex traits like HT. It allows young bulls and heifers to be selected on their GEBVs, thereby accelerating genetic gain [23]. Genomic applications enable the prediction of GEBVs for the selection candidates based on their genotypes [24,25]. An Australian breeding value for HT (HT ABV) in Holstein and Jersey cows and bulls based on the magnitude of the decline of milk, fat, and protein yield per unit increase in THI has been developed [24] and was incorporated into Australian national genetic evaluations in December 2017 [20]. HT ABV allows farmers to identify animals with a greater ability to tolerate hot, humid conditions with less impact on milk production [26,27].

In Australia, DataGene (www.datagene.com.au (accessed on 24 October 2022)) uses genomic technologies to estimate breeding values for dairy cattle. DataGene is an independent and industry-owned organization responsible for driving genetic gain and herd improvement in the Australian dairy industry and is an initiative of the dairy industry in Australia. Genetics contributes about 30% of production gains on Australian dairy farms, and DataGene’s genetic evaluation system underpins these gains. A key goal is to increase the number of farmers breeding replacements from good bulls and using Australian breeding values (ABVs) and indices to make breeding decisions [28]. Table 1 provides a description of the main dairy cattle production selection indices and ABVs used in the present study and their significance for the dairy farmer. Heat tolerance Australian breeding values (HT ABVs) [26] provide estimates of the genetic merit for the performance of dairy cows and bulls under HS. Genomic selection for HT should increase the resilience and welfare of dairy herds worldwide and the productivity of dairy farming in the future, given the expected increased incidence and duration of HS conditions. Therefore, this research aimed to study the association between HT phenotypic markers and genomic estimated breeding values for Australian dairy cattle production selection indices in a herd of Holstein Friesian cows.

## 2. Materials and Methods

The present experiment was approved by the University of Melbourne Faculty of Veterinary and Agricultural Science (FVAS) Animal Ethics Committee (AEC ID 1814645.1). The Australian dairy population is an ideal study model for HT because animals are predominantly kept outdoors in pastures where they experience the direct effects of weather elements [9].

### 2.1. Data Collection

The location of this study and the experimental animals, and its management and and data collection procedures, have been previously published [6]. Briefly, we recorded weekly physiological and milk production data of 150 Holstein Friesian dairy cows kept at the University of Melbourne, Robotic Dairy Farm, at Dookie, North Victoria, Australia, between 1st December 2018 and 28th February 2019, with ambient temperatures ranging from 18 to 42 °C and relative humidity of 25–75%. Phenotypic data collected included respiratory rate (RR), panting score (PS), and surface body temperature (SBT). Respiration rate was recorded via time in seconds taken for standing cows to make five flank movements (as the animal inhales and exhales with each breadth) [29] and calculated as respiration rate/minute. Animals were also observed for signs of drooling and/or open-mouth panting, and these data were used to determine the PSs of all cows [30]. The surface body temperature of the cows was determined non-invasively using an infrared thermal camera, FLIR T1050sc 28 [31]. Daily milk production, milk quality (somatic cell count, milk fat, and protein %), cow weights, and concentrate intake were collected automatically by the robotic milking machine (Lely Automatic Milking System), identifying individual cows via Radio Frequency Identification (RFID) ear tags. Additionally, each cow is fitted with a transponder (Qwes-HR, Lely) that contains a rumination monitor. The rumination monitor uses a microphone to detect chewing sounds and differentiates between eating and rumination time. Based on their physiological (SBT, RR, and PS) and milk production data blocked by stage of lactation, we selected the 20 most thermo-susceptible (Group 1) and 20 most thermotolerant cows (Group 2) out of the experimental herd over the summer period for genotyping. To gauge the effect of age on the parameters studied, we categorised the cows into 3 age categories as follows (Category 1: <5 years; Category 2: 5–7 years; Category 3: >7 years).

### 2.2. Genotyping

Genotyping was performed by an Australian Genetics company, Zoetis (https://genetics.zoetis.com/Australia (accessed on 24 October 2022)), whilst the breeding values and indices of experimental cows were estimated by DataGene (www.datagene.com.au (accessed on 24 October 2022)). To achieve this objective, we collected hair samples from the tip of the tail of experimental cows following a standard sample collection protocol for genotyping provided by Zoetis. Zoetis [32] uses the genomic selection tool CLARIFIDE that enables the detection of superior dairy heifers from as early as birth. ABVs are the best estimate of a female’s genetic merit and measure the characteristics (traits) she is likely to pass on to her offspring. ABVs are available for more than 45 different traits. The most economically important ABVs are incorporated into the breeding indices: the Balanced Performance Index (BPI) and the Health Weighted Index (HWI). The reliability of a cow’s ABV depends on the quality and quantity of information provided by the herd recording systems. In general, the more information used to calculate an ABV, the more accurate it is and the higher its reliability. Genotyping animals provides a significant boost to the reliability of ABVs. For instance, as of April 2020, a genotyped heifer had an average ABV reliability of 78% compared to one that is not genotyped, whose ABV was based on the parent’s average of 39% [28].

### 2.3. Prediction of GEBVs for HT

Heat tolerance (HT) has a genomic-only breeding value and does not have conventional breeding values like many other traits. In the current study, GEBVs for HT were estimated following the method described in detail by Nguyen et al. [33]. Briefly, SNP effects for the decline in milk, fat, and protein with increasing heat stress were calculated using a random regression model, per Nguyen et al. [24,33]. These slopes were used as pseudo phenotypes for HT for sires. A prediction equation for HT was developed using the combination of these pseudo-phenotypes and genotypes as described in Nguyen et al. [24,33]. The breeding value for HT was then expressed to have a mean of 100 and a standard deviation of 5 [33]. The reliability of HT ABVg in genotyped Holstein bulls with no daughters in the reference set ranged from 16 to 54%, with a mean of 38% and a standard deviation of 7% [33].

### 2.4. Prediction of Australian GEBVs for Other Traits and Selection Indices

Prediction of GEBVs for the other traits and indices was based on the method described in detail by Nieuwhof et al. [34], which briefly consisted of four major steps as follows:Quality assurance of the genotype-evaluating call rate and genetrain scores for each marker in a batch, lack of variation in the X-chromosome for males, duplicates in a batch indicating sampling issues, and duplicate genotypes for different animals across batches, indicating monozygotic twins or clones (which may cause dependencies in the analysis), Hardy Weinberg equilibrium and genotype inconsistencies given the pedigree.Imputation of missing genotypes or genotypes failing to meet the minimum genetrain score.Estimation of Direct Genetic Values (DGVs) using BLUP (SNP BLUP) described as RR-BLUP [35], based on an assumption that SNP effects are random and the DGV for bull *i* called *g_i_* is defined as follows:
$$g_{i}=\sum_{k=1}^{p}x_{ik}\beta_{k}$$
where $x_{ik}$ is a vector describing the genotype of bull *i* for *p* SNPs and $\beta_{k}$ is a vector with k SNP effects. The SNP effects were found by solving:β = (X’X + Iλ)^−1^X’y
where y is the phenotype and X is the matrix with vectors $x_{ik}$ for all bulls. In this equation, λ is defined as $\sigma_{e}^{2}$/$\sigma_{g}^{2}$, with $\sigma_{g}^{2}$ the genetic variation captured by the SNPs, and $\sigma_{e}^{2}$ is the error.

iv.Blending was based on Harris and Johnson’s [36] estimation of genomic breeding values (GEBVs).

### 2.5. Statistical Analysis

The following analysis of variance statistical model was used to study the relative effects of age category and thermotolerance group on physiological and production parameters, as well as variation in predicted GEBVs using Genstat software Version 22 [37], and the results are presented in the Tables and Figures:$$y_{ijk}=\mu +a_{i}+t_{j}+at_{ij}+e_{ijk}$$
where $y_{ij}$ = the observed individual value.

$a_{i}$ = the effect of age in the $i^{th}$ age category (*i* = 1, 2, and 3).

$t_{j}$ = the effect of thermotolerance in the $j^{th}$ thermotolerance group (*j* = 1 and 2).

$at_{ij}$ = the interaction effect between the age category and thermotolerance group.

$e_{ijk}$ = the error or residual effect.

In addition, Pearson correlation coefficients between GEBVs of the studied economic traits of dairy cattle (BPI, ASI, HWI, TWI, milk, protein, fat, feed saved, and fertility) were also determined.

## 3. Results

### 3.1. Variation in Physiological and Production Performance by Relative Thermotolerance

The average physiological response to HS and milk production performance of the experimental cows based on their relative thermotolerance classification are shown in Table 2.

### 3.2. Variation in Genomic Estimated Breeding Values (GEBVs) by Relative Thermotolerance

There was no difference (*p* > 0.05) in most of the GEBVs (Table 3) between the phenotypically thermotolerant and thermos-susceptible cow groups. However, numerically thermotolerant groups were inferior in most of the studied GEBVs except, as expected, the heat tolerance (HT) and feed saved (feed saved when a cow is smaller and needs less feed for maintenance and has a lower residual feed intake).

### 3.3. Variation in GEBVs of Selection Indices by Age Group

The Balanced Performance Index (BPI), Health Weighted Index (HWI), Type Weighted Index (TWI), and Australian Selection Index (ASI) blend production, type, and health traits for maximum profit. In the present study, younger cows were superior to older cows in BPI, ASI, HWI, TWI, milk fat, and milk protein GEBVs, whilst the reverse was true for HT (Table 4). Across age, there was not much variation in temperament and fertility GEBVs of the experimental cows, but some variation was obtained in mastitis resistance and especially feed saved GEBVs (Figure 1).

### 3.4. Association of GEBVs of Economic Traits

In terms of associations between GEBVs of the studied traits presented in Table 5, we observed highly significant (*p* ≤ 0.01) negative correlations (from −0.28 to −0.74) between HT and current dairy industry economic performance indices (BPI, ASI, HWI, TWI, milk, milk protein, and milk fat content) with positive correlations (*p* ≤ 0.05) between HT and feed saved (+0.45) and fertility (+0.25). Variations between age and HT with some of the studied GEBVs are shown in Figure 2 and Figure 3. Additionally, we found a large variation in feed saved and HT GEBVs (Figure 4 and Figure 5) and an effect of age on the BPI GEBV (Figure 4). Younger dairy cows had better temperament than older cows, which were also superior in HT GEBV (Figure 5). Finally, as shown in Figure 5, the yearlings were inferior to the older cows in feed saved GEBVs.

## 4. Discussion

As expected, the relatively more thermotolerant cows had higher (*p* ≤ 0.05) milk production, concentrate intake, and rumination time compared to the thermos-susceptible group, which were more impacted by the negative effects of HS. However, we did not find any significant differences between the two groups in terms of the physiological parameters, which could be partly attributed to the sample size. In terms of panting scores, experimental cows were, on average, classified as 2, meaning they demonstrated fast panting with drooling with no open mouth [30]. Findings of GEBVs of the HWI and TWI in the present study suggest that experimental cows were, on average, superior in the HWI (62) compared to the TWI (34). This implies that management is putting some emphasis on traits such as fertility, mastitis resistance, and feed saved (efficiency) in selection decisions. This should help make the dairy herd more resilient and, with their efficiency in feed saved, make the enterprise more profitable and less harmful to the environment. The average ASI GEBV of the 25 obtained in the current study indicates that there is more room for improvement in selecting bulls to produce daughters with the most profitable combination of protein, fat, and milk production. Even though the Dookie Robotic Dairy Farm is both a research and a commercial facility, management needs to review its breeding goals and amend them accordingly to be able to select more profitable bulls for the insemination of cows. The heat tolerance Australian breeding value (HT GEBV) allows farmers to identify animals with a greater ability to tolerate hot, humid conditions with less impact on milk production, and is expressed as a percentage with a base of 100 [38]. Dairy cows at the University of Melbourne’s Dookie Robotic Dairy Farm had HT GEBVs ranging from +93 to +112, with an average of +103. However, on average, there was no significant difference in HT GEBVs of the thermotolerant and thermos-susceptible groups. This means that even the cow with the best genetics in terms of heat tolerance is just 12% above the Australian dairy herd, with some 7% less tolerant than the average. This is confirmed by the negative association between HT and all three economic selection indices (BPI, ASI, and HWI), confirming that some of our experimental cows are genetically prone to heat stress. Although the heritability of HT is moderate at, on average, 0.19, genetic selection is expected to achieve significant progress [25]. Selecting breeding bulls based on HT has clearly not been given much weight by management in the past, and future selection decisions should consider help overcome this challenge to make the dairy herd more thermotolerant given the projections of extremely high global temperatures and heat waves [3].

Therefore, in the future, it will be important to consider HT as a critical component of the breeding objectives of dairy farmers in Australia. Additionally, HT is favourably correlated with fertility and unfavourably with production, meaning that high selection pressure for HT may improve fertility but compromise production. Significant (*p* ≤ 0.05) negative correlations (−0.39 to −0.69) were observed between HT and current dairy industry economic indices (BPI, TWI, ASI, and milk production). In contrast, positive correlations were recorded between HT and feed saved (+0.44) and fertility (+0.27), in line with previous findings [24]. These findings confirm that HT EBV is not currently included in the BPI [17,24]. In the future, dairy cattle breeders may want to choose bulls with high HT and BPI GEBVs to ameliorate the high environmental heat load, particularly during the summer months.

Another interesting finding was the large variation in feed saved and HT GEBVs, indicating the potential for selecting cows with these traits. Feed saved is defined as the amount of feed that is saved through improved metabolic efficiency and reduced maintenance requirements [26]. Feed saved, an index of feed conversion efficiency, is an important breeding goal because feed is a major cost variable in livestock production systems. Feed saved ABV has been available in Australia and included in the national selection indices since April 2015. The present study’s findings indicate that the relatively younger experimental cows are less likely to be efficient in feed utilisation due to the negative correlation between feed saved and milk yield. Feed saved ABV allows one to breed cows with reduced maintenance requirements for the same amount of milk produced. Feed saved is included in each of the three indices (BPI, HWI, TWI), with the highest weighting in the HWI, and is expressed in kilograms of dry matter of feed saved per cow per year, more or less than the average of zero. A positive number represents feed saved; a negative number represents extra feed consumed, which means that a lot needs to be improved with an average feed saved GEBV of +26 in the current study. To improve feed efficiency in this herd, management will need to select bulls in the future with positive feed saved GEBVs. Heat-stressed animals consume less feed and produce less milk, and, therefore, there is a need for strategies to mitigate the impacts of HS on animal performance [25]. Consequentially, animals superior in feed saved GEBVs should be more HT and adaptive to adverse effects of heat stress on foetal and mammary development arising from disruptions in placental function during pregnancy [11].

We also investigated the relationship between GEBVs of current dairy cattle performance indices and found a wide variation in feed saved GEBVs (from −93 to +130) across age, indicating a potential for their selection. Such genetic variation provides flexibility to adapt to the changing environment and enhances the survival of the population over time [39]. Therefore, identifying and selecting animals that are thermotolerant is a viable alternative for reducing the adverse effects of HS on dairy cattle performance [40]. Moreover, the relatively thermotolerant experimental cows had a somewhat higher GEBV for feed saved and fat% but lower milk production potential. The breeding goal of Australian dairy breeding is clearly seen in the effect of age on the BPI and temperament GEBVs of dairy cows, with younger dairy cows ranking better in these traits. On the other hand, older cows seem to be genetically more heat tolerant than younger cows, which may be due to the negative antagonism between heat tolerance and milk yield; thus, with the current breeding objective increasing milk yields, younger cows are inferior in terms of the genetics for HT, and this needs to be corrected in future breeding programmes. Overall, future breeding goals should focus on climate-smart productive dairy cattle in a sustainable environment. In this regard, relationships between new indices, such as the sustainability index [41], which allows farmers to fast-track genetic gain for reduced greenhouse gas emission (GHG) intensity, and the GEBVs and selection indices studied here should be explored.

## 5. Conclusions

In the present study, we tested the reliability of GEBVs for HT under Australian natural summer by recording phenotypic data on Holstein Friesian lactating cows. In terms of the phenotypically relatively thermotolerant and thermos-susceptible groups, although we did not find significant differences between them due probably to a limitation in sample size, as expected, the thermo-susceptible group recorded relatively higher GEBVs for the BPI, ASI, HWI, TWI, milk production, milk protein, and milk fat. However, in line with their physiological response to heat stress, the thermotolerant group had a relatively higher GEBV for HT than the thermos-susceptible group. Genomic estimated breeding values of the BPI, TWI, ASI, and HWI are superior in younger cows, whilst older cows were superior in fertility, feed saved (FS), and heat tolerance (HT). In general, based on the estimated GEBVs, most of our experimental cows at the University of Melbourne’s Dookie Robotic Dairy are daughters of bulls that were selected based on superior milk production performance and not so much on their feed efficiency, resistance to mastitis, or their ability to survive prolonged heat wave events. Positive associations between HT and FS, as well as HT and fertility GEBVs, indicate that selection for HT may help improve cow efficiency and sustainability of production under hot summer conditions. This calls for more climate-smart breeding goals in the future by increasing the weighting on HT in order to breed for more resilient and production-efficient dairy herds. Such an effective strategy to breed high-producing and adaptive ruminants to feed the growing world population under changing climatic conditions will greatly boost sustainable ruminant livestock production. Therefore, we recommend a more extensive study involving a larger number of lactating cows across multiple farms to confirm these associations to adopt and incorporate HT breeding values into various dairy economic performance indices for the selection of dairy cattle to improve efficiency, fertility, and thermotolerance of future dairy cows in a sustainable environment.

## Figures and Tables

**Figure 1 animals-13-02259-f001:**
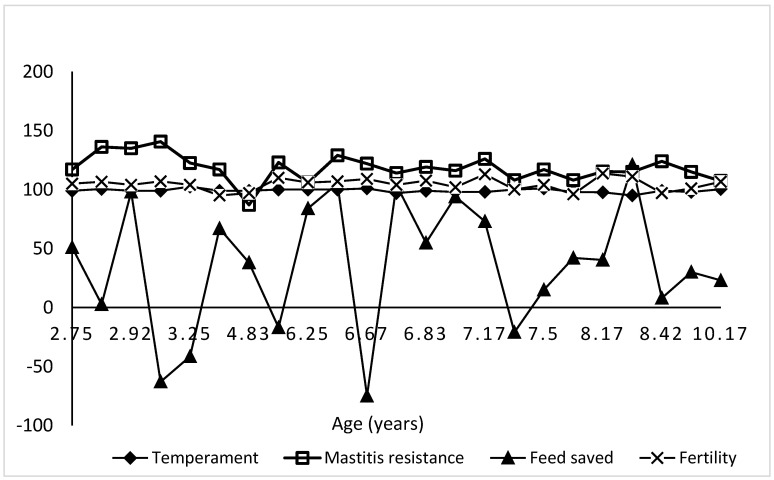
Variation of temperament, mastitis resistance, feed saved, and fertility GEBVs with age in Australian dairy cattle.

**Figure 2 animals-13-02259-f002:**
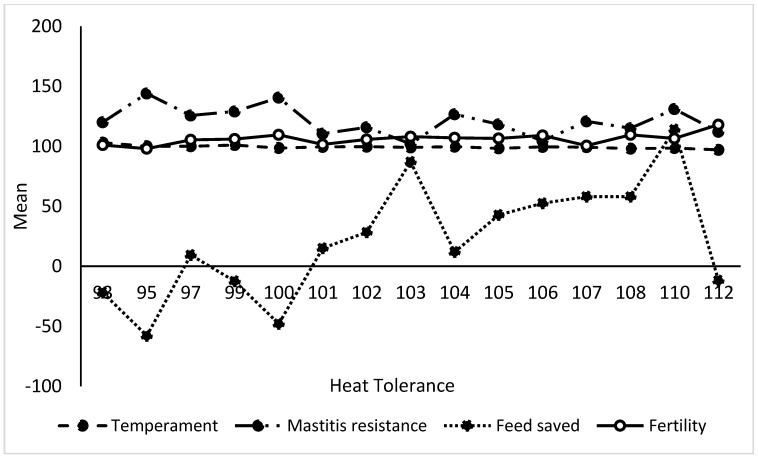
Variation of temperament, mastitis resistance, feed saved, and fertility GEBVs with heat tolerance GEBVs of Holstein dairy cows.

**Figure 3 animals-13-02259-f003:**
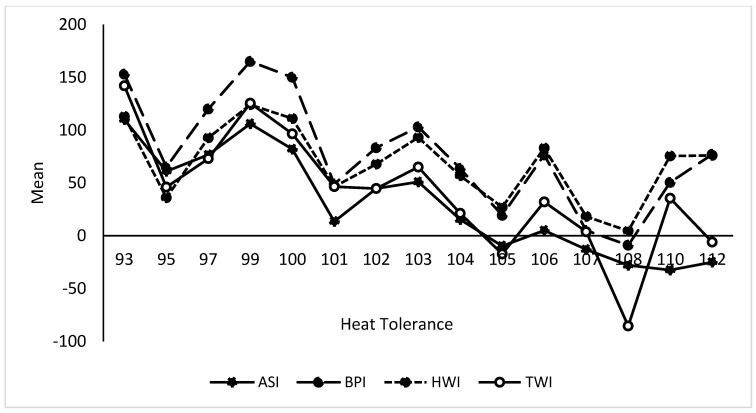
Variation in the BPI, ASI, HWI, and TWI GEBV with the heat tolerance performance index.

**Figure 4 animals-13-02259-f004:**
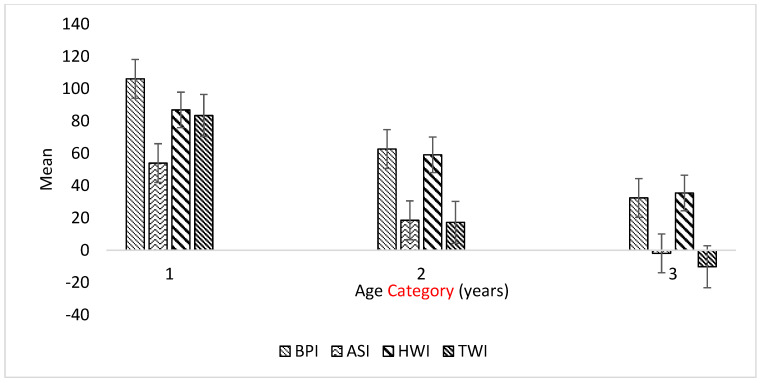
Effect of age on the mean GEBVs of the BPI (*p* < 0.05), ASI (*p* < 0.01), HWI (*p* = 0.09), and TWI (*p* < 0.01) in Australian dairy cattle. Category 1: <5 years; Category 2: 5–7 years; Category 3: >7 years. Interaction effects of the thermotolerance group and age category were not significant (*p* > 0.05).

**Figure 5 animals-13-02259-f005:**
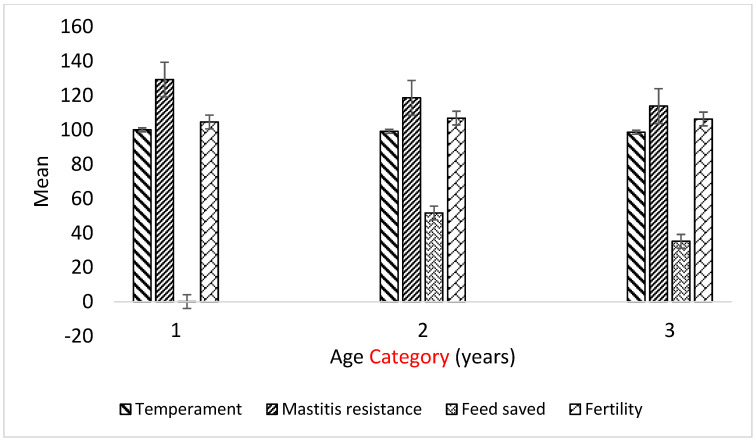
The effect of age on mean GEBVs of temperament (*p* < 0.05), mastitis resistance (*p* < 0.01), feed saved (*p* = 0.06), and fertility (*p* = 0.5) in Australian dairy cattle. Category 1: <5 years; Category 2: 5–7 years; Category 3: >7 years. Interaction effects of the thermotolerance group and age category were not significant (*p* > 0.05).

**Table 1 animals-13-02259-t001:** Description of estimated Australian dairy performance indices and breeding values.

Parameter	Description	Remarks
Balanced Performance Index (BPI)	The Balanced Performance Index (BPI) is an economic index that balances the economic contribution of production, health and fertility, type, workability, and feed efficiency. The updated BPI applies greater emphasis to health by adding in survival and mastitis resistance.	The BPI identifies bulls and cows that combine traits that are important to profit. Farmers can track this in their genetic progress report and make appropriate and timely breeding decisions.
Health Weighted Index (HWI)	The Health Weighted Index (HWI) allows farmers to fast-track traits such as fertility, mastitis resistance, and feed saved (efficiency).	The HWI puts the greatest emphasis on health and fertility, with production secondary.
Type Weighted Index (TWI)	The Type Weighted Index (TWI) allowed farmers to fine-tune type traits to make a good herd even better.	Currently, the TWI has been replaced by good bulls guide tables.
Australian Selection Index (ASI)	The ASI is a production-based index that ranks animals (bulls or females) on their ability to produce daughters with the most profitable combination of protein, fat, and milk production. Traits are weighted according to the way Australian dairy farmers are paid for their milk (fat + protein − volume). The ASI is expressed in dollars. An ASI of 200 means this animal is AUD200 per year more profitable from production than average.	The ASI is included in all three indices (the BPI, HWI, and sustainability indices) with the highest waiting on the sustainability index. For example, if an animal has an ASI of 200, then that is the contribution to production. If that same animal has a BPI of 300, then BPI 300 = ASI 200 + 100 from non-production.
Feed Saved (FS) ABV	The feed saved ABV allows one to breed cows with reduced maintenance requirements for the same amount of milk produced. It is expressed in kilograms of dry matter of feed saved per cow per year more or less than the average of zero. A positive number represents feed saved; a negative number represents extra feed consumed. In genotyped Holsteins, feed saved ABV utilises maintenance feed requirements predicted from type traits and Residual Feed Intake (RFI). Reliability is a measure of confidence in an ABV. The reliability of an animal’s breeding values improves with age as more information becomes available; for example, genomics, daughters’ performance records, and herd test results.	To improve feed efficiency in your herd, select animals with a feed saved ABV greater than zero. Feed saved is a moderately heritable trait (20–30%), which means that selection for feed saved will make a difference. An updated model for the feed saved ABV was implemented in November 2020, resulting in improved reliability (42–45%). For Holstein bulls, this represented an 11% improvement in reliability.
Heat Tolerance (HT) ABV	HT ABV allows farmers to identify animals with a greater ability to tolerate hot, humid conditions with less impact on milk production. It is expressed as a percentage, with a base of 100. An animal with an ABV of 105 is 5% more tolerant to hot, humid conditions than the average, and its drop in production will be 5% less than the average. On the other hand, an ABV of 95 means the animal is 5% less tolerant to hot, humid conditions than the average and its drop in production under heat stress is 5% more than the average.	To improve heat tolerance in your herd, select animals with a heat tolerance ABV of greater than 100. Allow for the lower reliability (36–38%) of the heat tolerance ABV by using a team of bulls. Reliability for HT ABV is expected to increase as more records become available.

Source: DataGene [28].

**Table 2 animals-13-02259-t002:** Mean (± SD) physiological and milk production data of the experimental cows.

Parameter	Group 1 (Thermo-Susceptible)	Group 2 (Thermotolerant)
Respiration rate (breadths min^−1^) ^#^	91.8 ± 34.7 (303) *	90.1 ± 32.1 (313)
Panting score ^λ^	2.0 ± 0.8 (307)	1.9 ± 0.8 (317)
Daily milk production (kg/d)	21.3 ± 5.6 ^b^ (341)	30.0 ± 6.9 ^a^ (340)
Fat %	4.4 ± 0.9 ^a^ (340)	3.9 ± 0.6 ^b^ (313)
Protein %	3.2 ± 0.3 ^a^ (340)	3.0 ± 0.2 ^b^ (340)
Concentrate intake (kg/d)	5.3 ± 1.8 ^b^ (322)	6.2 ± 1.6 ^a^ (320)
Rumination time (mins)	399.4 ± 108 ^b^ (320)	445.9 ± 108.5 ^a^ (320)
Residual feed (kg/d)	1.1 ± 0.2 ^a^ (322)	0.7 ± 0.8 ^b^ (322)

* Number of observations in brackets; ^#^ within rows means followed by different superscripts are significantly (*p* ≤ 0.05) different. ^λ^ Based on the scale used by Gaughan et al. [30].

**Table 3 animals-13-02259-t003:** Variation of mean GEBVs (± SD) of the selected traits with heat tolerance ability.

	Thermo-Susceptible Group (*n* = 19)	Thermotolerant Group (*n* = 20)	Herd Average
*n* (sample size)	19 *	20	39.0
BPI	75.7 ± 19.2	63.2 ± 16.5	69.3
ASI	32.0 ± 12.5	19.0 ± 11.9	25.3
HWI	65.6 ± 15.3	58.4 ± 13.3	61.9
TWI	37.8 ± 21.5	29.5 ± 14.3	33.5
Milk	86.7 ± 68.7	−14.1 ± 91.2	35.0
Milk protein	4.4 ± 1.8	2.4 ± 1.6	3.3
Milk fat	6.1 ± 1.6	0.25 ± 2.6	3.1
HT	102.4 ± 0.95	104.1 ± 0.93	103.2
Feed saved	20.8 ± 12.0	31.5 ± 14.3	26.28
Fertility	106.0 ± 1.05	105.8 ± 1.38	105.9

* Bad genotyping results of one sample of the most thermo-susceptible group were discarded.

**Table 4 animals-13-02259-t004:** Variation in average GEBVs of the selected traits with the age of dairy cattle.

	Age Category			
	<5 Years	5–7 Years	>7 Years	Total/Overall
*n*	15	11	13	39
BPI	106.1 ^a^ ± 21.3	62.6 ^ab^ ± 17.6	32.4 ^b^ ± 20.1	69.26
ASI	53.9 ^a^ ± 15.6	18.5 ^ab^ ± 12.8	−1.9 ^b^ ± 10.4	25.33
HWI	86.9 ± 16.8	59.1 ± 13.1	35.5 ± 18.1	61.9
TWI	83.4 ^a^ ± 17.9	17.3 ^ab^ ± 20.1	−10.2 ^b^ ± 19.5	33.54
Milk	207.1 ^a^ ± 79.0	−175.3 ^b^ ± 131.1	14.4 ^ab^ ± 68.7	35.0
Milk protein	9.0 ^a^ ± 1.7	−0.18 ^b^ ± 2.0	−0.23 ^b^ ± 1.5	3.33
Milk fat	7.0 ± 3.0	1.2 ± 2.8	0.2 ± 2.2	3.10
HT	100.9 ± 1.2	103.9 ± 1.0	105.4 ± 0.80	103.2
Feed saved	−0.3 ± 15.04	51.7 ± 19.7	35.2 ± 11.0	26.28
Fertility	104.7 ± 1.43	106.91 ± 0.80	106.4 ± 1.91	105.9

GEBVs = genomic estimated breeding values; BPI = Balanced Performance Index; ASI = Australian Selection Index; HWI = Health Weighted Index; TWI = Type Weighted Index; HT = heat tolerance; *n* = sample size. Means with different superscripts differ significantly (*p* ≤ 0.05) within each row.

**Table 5 animals-13-02259-t005:** Pearson correlation coefficients between GEBVs of economic traits.

	BPI ^#^	ASI	HWI	TWI	Milk	Protein	Fat	FS	Fertility
ASI	0.80 **								
HWI	0.97 **	0.64 **							
TWI	0.95 **	0.77 **	0.92 **						
Milk	0.11	−0.05	−0.13	−0.02					
Protein	0.52 **	0.70 **	0.40 **	0.58 **	0.64 **				
Fat	0.61 **	0.80 **	0.46 **	0.56 **	−0.02	0.46 **			
FS	−0.30	−0.41 **	−0.13	−0.32 *	−0.29	−0.45 **	0.48 **		
Fertility	0.51 **	0.02	0.62 **	0.30	−0.28	0.18	0.04	0.03	
HT	−0.43 **	−0.70 **	−0.28	−0.45 **	−0.33 *	−0.74 **	−0.59 **	0.45 **	0.25

^#^ GEBVs = genomic estimated breeding values; BPI = Balanced Performance Index; ASI = Australian Selection Index; HWI = Health Weighted Index; TWI = Type Weighted Index; HT = heat tolerance; FS = feed saved; *n* = sample size. ** = *p* < 0.01; * *p* < 0.05; sample size (*n* = 39 cows).

## Data Availability

The data presented in this study are available on request from the corresponding author. The data are not publicly available due to institutional restrictions.

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
