# Peer review of "Association of Phenotypic Markers of Heat Tolerance with Australian Genomic Estimated Breeding Values and Dairy Cattle Selection Indices"

_animals, 2023, doi:10.3390/ani13142259_

Round 1
Reviewer 1 Report
animals-2423406
Title:Association of Phenotypic Markers of Heat Tolerance with Australian Genomic Estimated Breeding Values and Dairy Cattle Selection Indices
Authors: R. Osei-Amponsah , Frank R Dunshea , Brian J Leury , Archana Abhijith , Surinder S Chauhan
Major concerns: It is an interesting report, however, there are some problems to be published.
1. Authors used some selection indices, Balanced Performance Index (BPI), Health Weighted Index (HWI), Type Weighted Index (TWI) and Australian Selection Index (ASI) without explanations of them.
Even authors know these indices very well, they had better to explain more details of these indices, otherwise, many readers do not understand the results, e.g. Figures and Table 4, well.
2. Mathematical models: Authors commented age effect on some traits among selected group (n=20+19) by heat tolerance ability.
Why authors did not apply any mathematical models to evaluate age and group effects separately.
Reviewer 2 Report
This paper is aimed to investigate the association of markers of heat tolerance with genomic estimated breeding values and dairy cattle selection indices. The authors used a real dataset which is valuable. However, there are some minor and major concerns which are summarized below:
Major:
1. Method of estimation of GEBVs must be described. The authors only mentioned that genomic breeding values are estimated by Zoetis. It is important to know that GEBVs of all of those traits are estimated using only one method or more. For instance, GEBVs of daily milk might be estimated using the GBLUP while GEBVs of Milk Fat might be estimated using ssGBLUP.
2. The ranges of GEBVs reliabilities for those samples used in this study must be mentioned. I wonder if the authors consider it in the sample collection.
3. Authors must add statistical models used for analyses in this study.
4. I wonder why the authors considered only genomic breeding values. Why they did not test estimated breeding values (EBVs) using traditional BLUP? Also, they could use de-regressed-EBVs.
5. How many individuals were considered for calculation of Pearson correlation coefficients among GEBVs of different economic traits?
Minor:
1. A “)” must be placed in line 97 or 98.
2. In Table 1, I guess “32.1” as the standard deviation of fat is not true. Also, replace “+” with “±” if those numbers are standard deviations.
3. In line 258, the “FS” is repeated in “Positive associations between HT and FS and FS and …”.
Reviewer 3 Report
REVISION TO MANUSCRIPT ASSOCIATION OF PHENOTYPIC MARKERS OF HEAT TOLERANCE WITH AUSTRALIAN GENOMIC ESTIMATED BREEDING VALUES AND DAIRY CATTLE SELECTION INDICES 4
ANIMALS-2423406
Mayor comments:
In section 2. Materials and Methods.
From line 108 to 109. Authors mention they select 20 most thermotolerant and 20 most thermos susceptible cows. In table 1 named Group 1 and Group 2.
From line 110 to line 111. Authors mention that cows grouped by age named as Group 1, Group 2, and Group 3.
At this point, the manuscripts potentially can be complicated to discriminant between both Group 1 and Group 2.
From line 113. Authors in Table 1 mention “# within rows means followed by different superscripts are 114 significantly (p ≤ 0.05) different” But there is any statistical analyses described in the section.
In addition, authors mention “# Based on the scale used by Gaughan et al. [28].”Nevertheless, in the text where authors describe PS and RR, there is any mention regarding this scale.
Please add details in order to clarify all the steps in Materials and Methods section.
From line 116 to line 124. Here the authors need to clarify this entire paragraph. Here authors must include all the information necessary to understand results and discussion. At this moments; look like a black box.
I would suggest to authors that after incorporate the mayor comments into a newest version of the manuscript; please update and get order results and discussion sections.
The conclusions must be supported by results in this study.
Round 2
Reviewer 1 Report
accepted
Author Response
Thank you for the interest shown in our work and for providing valuable suggestions to improve on the quality of our manuscript.
Thank you for accepting the revised version of our manuscript and recommending it for publication.
Reviewer 2 Report
Some of my comments were applied in the revised version. However, there are still some of my concerns:
1. Statistical method of estimation of GEBVs must be described. The authors only mentioned that genomic breeding values are estimated by Zoetis. It is important to know that GEBVs of all those traits are estimated using only one method or more.
2. Authors did not provide a convincing reason in order to use GEBVs instead of EBVs and its deregressed in the manuscript.
Also regarding the provided statistical model to isolate the relative effects of age category and thermotolerance group on GEBVs:
I am not sure if this model is conceptually suitable for isolating the relative effects of age category and thermotolerance group on GEBVs or not. In breeding companies, most of the phenotypic records are already adjusted based on some fixed effects (such as age category, sex and etc.) before the estimation of GEBVs.
Author Response
Thank you for the interest shown in our work and for providing valuable suggestions to improve on the quality of our manuscript.
Please see the attachment.

Reviewer 3 Report
Great job!
Great job!
Author Response

(The authors gave the same response as above.)
